# Experiences of Racial and Ethnic Discrimination Are Associated with Food Insecurity and Poor Health

**DOI:** 10.3390/ijerph16224369

**Published:** 2019-11-08

**Authors:** Pam Phojanakong, Emily Brown Weida, Gabriella Grimaldi, Félice Lê-Scherban, Mariana Chilton

**Affiliations:** 1Department of Epidemiology and Biostatistics, Dornsife School of Public Health at Drexel University, Philadelphia, PA 19104, USA; pp364@drexel.edu (P.P.); fzl23@drexel.edu (F.L.-S.); 2Department of Health Management and Policy, Dornsife School of Public Health at Drexel University, Philadelphia, PA 19104, USA; eab364@drexel.edu (E.B.W.); gg394@drexel.edu (G.G.)

**Keywords:** racism, discrimination, food insecurity, food security, depression, self-rated health, adverse childhood experiences (ACEs), trauma

## Abstract

This study examines the associations of mothers’ experiences of discrimination (EODs) with household food insecurity (HFI), physical health, and depressive symptoms, while taking into account the influence of mothers’ Adverse Childhood Experiences (ACEs) and public assistance participation. Mothers (N = 1372) of young children under age 4 who self-identified as Latinx, Non-Latinx Black/African American and Non-Latinx white answered questions for a cross-sectional survey in an emergency room in a large children’s hospital in Philadelphia between 2016 and 2018. Logistic regression was used to model associations of EODs in specific settings with HFI, depressive symptoms, and physical health. Compared to those without EODs, mothers with EODs from police/courts and in workplaces had higher odds of HFI, AOR =2.04 (95% CI: 1.44–2.89) and AOR = 1.57 (95% CI: 1.18–2.11), respectively. Among Latinx mothers, EODs in school were associated with nearly 60% higher odds of HFI and nearly 80% higher odds of depressive symptoms. Latinx and Black mothers with EODs in workplaces had higher odds of HFI (AOR = 1.76, 95% CI: 1.21–2.56 and AOR = 1.46, 95% CI: 1.05–2.36, respectively), compared to mothers without EODs. Discrimination is associated with HFI, depressive symptoms, and poor health. Public health interventions intended to improve food security and health may be only partially effective without simultaneously addressing racism and discrimination.

## 1. Introduction

Interpersonal experiences with perceived racism cause emotional and psychological harm resulting in trauma-related symptoms such as depression, disordered sleep, hyper vigilance, isolation, and overall poor health [1,2,3,4]. Additionally, systemic racism such as disparate sentencing laws based on race/ethnicity, housing discrimination, discrimination in health care, and disparities in school funding have also been found to contribute to poor health [5]. Taken together, discrimination in a variety of settings, whether interpersonal or experienced through the systems that disproportionately affect the functions of everyday living, is likely at the root of racial and ethnic disparities in health and their social determinants, such as household food insecurity (HFI). For example, in 2018, 21.2% of African American or Black households and 16.2% of Hispanic households reported food insecurity, compared to 8.1% of white households [6]. Similarly, in a recent nationwide study, 15% of Black adults and 11% of Hispanic adults reported being in fair or poor physical health, compared to 9% of white adults [7]. Black individuals are also more likely than non-Latinx whites to report depressive symptoms, though reports comparing differences between Latinx and non-Latinx whites are mixed [8].

While there is ample evidence of persistent racial and ethnic health disparities, there are methodological weaknesses in the conventional public health approach—namely, its emphasis on individual mechanisms rather than the subtle and systemic ways racism shapes access to opportunities in education, employment, housing and neighborhood resources and affects individual and collective physical, mental, and spiritual health [9]. Critical Race Theory calls for attention to equity and critical self-reflection from investigators and public health practitioners; this approach demands that we take a hard look at previous methods and improve current methods to address issues of systemic racism [10]. Using the lens of Critical Race Theory and given (1) the paucity of studies that investigate health impacts of discrimination in specific settings among mothers of young children and (2) the limited research that simultaneously investigates the impact of discrimination among multiple racial/ethnic groups [11], the current study sought to examine the associations between race and ethnicity-based discrimination and multiple health outcomes. More specifically, we hypothesized that among mothers of children under age 4, lifetime experiences of discrimination (EODs) in a variety of settings, such as by police and courts, in public, at work and in school [12], would be associated with HFI, caregiver depressive symptoms, and fair/poor physical health, and that there would be differential associations by race and ethnicity. Understanding that EODs are a form of adversity linked to trauma-related symptoms, we also assessed EOD relations to these outcomes while taking into account exposure to adverse childhood experiences (ACEs), which include experiences with abuse, neglect, and household adversity such as witnessing violence or having a parent in prison. ACEs have long been associated with major negative health outcomes, such as cardiovascular disease, diabetes, and depression [13,14,15], and have also shown significant association with household food insecurity [16,17,18]. We hypothesized that EODs would be associated with poorer health outcomes even after accounting for past exposure to ACEs. Finally, given evidence that nutrition assistance programs help reduce household food insecurity and related poor health [19,20,21], we hypothesized that public assistance participation may attenuate or eliminate these associations.

## 2. Materials and Methods

### 2.1. Study Population

Participants were from an original sample of 1707 parents (primary caregivers that are biological parents, non-biological parents, grandparents or legal guardians) from a large ongoing cross-sectional study among parents of young children from December 2016 to December 2018. Eligible participants were a convenience sample of parents of children under age 4 years who were not in critical condition (designated in the electronic medical record by triage staff or admitted for injury from assault) in the Emergency Department at a large children’s hospital in Philadelphia during daytime business hours. Trained interviewers approached eligible parents while they were waiting to be seen by a nurse or physician and otherwise unoccupied to investigate associations between food insecurity, health, public assistance participation and employment. Once the interview was complete, participants verbally consented to answer supplemental questionnaires covering Experiences of Discrimination (EODs) and Adverse Childhood Experiences (ACEs). Interviews were conducted in English or Spanish. The Drexel University Institutional Review Board approved this study.

Of 1707 eligible parents who completed the initial interview, 1643 parents completed the full questionnaire. Because race and ethnicity are measured in reference to biological mothers only, 160 survey participants (non-biological mothers) were excluded from the analysis. While 1483 mothers were eligible for analysis, 109 declined to participate in the ACE and EOD survey interview, leaving 1372 mothers in the final analytic sample. The 109 mothers who declined to participate were on average slightly older compared to mothers who completed the survey (28.6 years vs. 27.4 years, Mann–Whitney U test *p* = 0.03). There were no missing data. There were no other significant differences by education, employment, race and ethnicity, physical and mental health status, adverse childhood experiences, insurance and benefits receipt, or food security.

### 2.2. Measurements

Mothers of children under 4 years who were not in critical distress were approached in the Emergency Department by trained interviewers to administer the survey and obtain other information from the medical record. Participants were provided with a laptop and headphones to answer the ACE and EODs measured using Audio Computer-Assisted Self-Interview (ACASI) software. The participant could also request to have the survey verbally administered, as all other questions in the survey were verbally administered. Further details of data collection procedures can be found in a previous publication [16]. The survey covered demographic and economic characteristics, including age, self-identified race/ethnicity, educational attainment, marital status, and public assistance participation. Receipt of public assistance (yes/no) was defined as the caregiver reporting currently receiving any of the following benefits: Temporary Assistance for Needy Families (TANF), Supplemental Nutritional Assistance Program (SNAP), Special Supplemental Nutrition Program for Women, Infants, and Children (WIC), or housing subsidies.

Household food security status was assessed using the Household Food Security Survey Module (HFSSM), a validated 18-item survey tool [22], which has been rigorously studied for reliability and internal consistency showing Cronbach’s α coefficients ranging from 0.73 to 0.95 [23]. In accordance with HFSSM categories, household food security is classified into high, marginal, low or very low, as a function of self-reported access to food and diet quality. In keeping with existent literature, mothers were further classified as *food secure*, which included high and marginal food security, indicating little or no food access problems, and *food insecure*, which included low and very low food security, indicating problems ranging from worry that food would run out before there was money to buy more, to reduced diet quality, to reduced food intake and disrupted eating patterns.

Maternal health status was measured using the Self-Rated Health question adapted from the National Health and Nutritional Examination Survey [24]. Mothers were asked: “In general, would you say your own physical health is excellent, good, fair, or poor?” The measure was dichotomized to excellent or good versus fair or poor. Depressive symptoms were assessed using a three-item maternal depressive symptoms screening tool validated for use in pediatric clinics that measures feelings of depression over multiple days in the past week, at least two weeks in the past year, and at least two years over the lifespan [25]. Depressive symptoms were indicated by affirmative response to at least two questions.

The Adverse Childhood Experiences (ACEs) Scale is a retrospective 10 question survey covering experiences of abuse (physical, emotional, and sexual), physical and emotional neglect, and household dysfunction, including parental separation or divorce, exposure to domestic violence, substance abuse, mental illness, and incarceration of a household member, before the age of 18 [14]. This measure has high internal consistency and construct validity with a Rasch infit mean square fit between 0.7 and 1.2 [26] and comparative fit index of 0.98 [27], and strong correlation with childhood trauma as well as health outcomes later in life [28]. Cumulative scores were calculated for mothers based on number of affirmative responses, in keeping with literature that recognizes significance of cumulative ACEs along a dose-response continuum [28]. As previous studies have found that the threshold of four or more ACEs is indicative of greater risk for later negative outcomes, the total score was then categorized into 0, 1 to 3, and 4 or more ACEs for analyses [14,29].

Exposure to racial and ethnic discrimination was measured using the validated 9-item self-reported Experiences of Discrimination (EODs) instrument [12,30]. One new question meant to capture the context of applying for benefits was added to the survey. This measure asks mothers whether they have “been prevented from doing something, having been hassled, or having been made to feel inferior because of [their] race, ethnicity, or color,” and about the frequency of experiences in various settings. The settings are: at school; getting hired or getting a job; at work; getting housing; getting medical care; getting service in a store or restaurant; getting credit, bank loans, or a mortgage; on the street or in a public setting; and from the police or in the courts. In order to address another common system that families might encounter; this study added the setting of applying for public assistance. Negative responses were scored as 0; affirmative responses as 1. While all EODs are potentially impactful, only the most commonly reported EODs (over 10% affirmative) were used to model the associations between discrimination, food insecurity and caregiver health for greater statistical power. Also, for simplicity in presented models, individual EODs that could be combined based on similarity of setting were collapsed into broader EODs categories. EODs in getting services in a store or restaurant and EODs in public or on the street were combined and categorized as EODs in public places. Similarly, EODs in getting a job or hired and EODs at work were combined and categorized as EODs in hiring and workplace.

### 2.3. Statistical Methods

Pearson’s chi-squared test was used to test differences in the distribution of EODs by self-reported race and ethnicity. Logistic regression was used to model associations of EODs with HFI, maternal physical health, and depressive symptoms. Each type of EODs was modelled separately with each outcome. Three sequentially adjusted models were used to estimate associations of EODs with the outcomes. Model 1 included known confounders: maternal age, race/ethnicity, education level, employment status, public health insurance, marital status, household size, place of mother’s birth, and, for those born outside the US and Puerto Rico, length of time in US (≥10 years living the United States vs. ≤10 years living in the United States). Length of time is accounted for because immigrant parents’ HFI and health differ in relation to length of time in the US [31]. In order to test associations of EODs with the outcomes independent of ACEs, model 2 included caregiver ACEs in addition to the variables from model 1. In order to descriptively assess whether associations of EODs with the outcomes might be attenuated by receipt of public assistance, which are known protective factors [32], model 3 included the indicator for current public assistance receipt in addition to variables from model 2. Finally, given previous evidence that discrimination has differential effects on people of different racial and ethnic backgrounds [33], we ran fully adjusted models (model 3) stratified by race/ethnicity. Test results were considered statistically significant at the α = 0.05 level. SAS version 9.4 was used for analyses.

## 3. Results

### 3.1. Participant Characteristics and Experiences of Discrimination

Mean caregiver age was 27.4 years; mean child age was 19.7 months (Table 1). Over half of the mothers (57.3%) self-identified as Latinx (Hispanic, Latino(a), or Spanish), 33.2 percent identified as Black non-Latinx, 7.9 percent identified as white non-Latinx, and 3.4 percent identified as another race/ethnicity or more than one race/ethnicity. The majority of mothers (85.8%) were born in the United States or Puerto Rico. Immigrant mothers were primarily from the Caribbean (7.1%), Mexico, and Central America (3.9%). Childhood adversity was prevalent with 35.6% of mothers reporting 1–3 ACEs and 12.0% of mothers reporting ≥4 ACEs. Eighty-two percent of mothers participated in public insurance (Medicare, Medicaid, military healthcare benefits) and most mothers reported receiving some form of public assistance. For example, 76.3% participated in SNAP, and 71.3% participated in WIC.

The most prevalent settings for discrimination experiences were in public or on the street (27.2%), stores and restaurants (20.1%), and at work (20.1%) (Table 2). Rates of experiences of discrimination were less than five percent in settings such as housing, medical care, and banking. Reports of experiences of discrimination were also low in applying for public assistance (6.9%, n = 94). EODs varied significantly by race/ethnicity across all settings except housing, medical care, and applying for public assistance. Apart from medical care and applying for public assistance benefits, rates of discrimination were highest among non-Latinx Black mothers.

### 3.2. Food Insecurity, Depressive Symptoms and Poor Health

As stated above, after collapsing some EOD items into broader categories, only the most common EOD items (prevalent among at least 10% of mothers: police or in courts, public places, hiring and workplace, school) were modeled in regression analyses. EODs were associated with HFI and health outcomes (Table 3). Adjusting for maternal age, race/ethnicity, educational attainment, years lived in the United States, insurance coverage, employment, marital status, and household size (Model 1), mothers who reported EODs from police or in courts had more than double the odds of HFI compared to mothers with no EODs from police or in courts (AOR = 2.54, 95% CI: 1.82–3.53). With further adjustment for ACEs in model 2, this association was attenuated, but remained statistically significant (Model 2 AOR = 2.06, 95% CI: 1.45–2.91). Subsequent adjustment for receipt of public assistance benefits in model 3 did not significantly change the strength of association (AOR = 2.04, 95% CI: 1.44–2.89). EODs in hiring and workplaces were similarly associated with higher odds of HFI (Model 3 AOR = 1.57, 95% CI: 1.18–2.11), as were EODs in school (Model 3 AOR = 1.44, 95%CI: 1.06–1.96). EODs in public places was associated with higher odds of HFI. However, once adjusted for ACEs and public assistance participation, this association was no longer statistically significant.

EODs in all settings were associated with elevated odds of depressive symptoms, ranging from AOR = 1.65 (95% CI: 1.16–2.33) for EODs in public places to AOR = 1.96 (95% CI: 1.30–2.97) for EODs in school. However, for EODs in all settings these associations were attenuated and no longer statistically significant after further adjustment for ACEs and public assistance benefits.

EODs from police or in courts, in public places, and in hiring and workplaces were all associated with higher odds of poor physical health status (Model 1: AOR = 1.47 [95% CI: 1.02–2.12], AOR = 1.59 [95% CI: 1.22–2.08], and AOR = 1.36 [95% CI: 1.02–1.83], respectively). The association of EODs in public places with poor physical health status was somewhat attenuated but remained statistically significant after adjusting for ACEs in model 2, (AOR = 1.36, 95% CI: 1.01–1.83). Further adjustment for receipt of public benefits in model 3 did not alter the association (AOR = 1.37, 95% CI: 1.02–1.84). Associations of physical health status with EODs from police or in courts and in hiring and workplaces were more strongly attenuated and no longer statistically significant after further adjustment for ACEs. In contrast, EODs in school was not associated with poor physical health in model 1 (AOR = 1.04, 95% CI: 0.99–1.09), but was associated with higher odds of poor physical health after further adjustment for ACEs in model 2 (AOR = 1.95, 95% CI: 1.28–2.97). This association was somewhat attenuated after additional adjustment for receipt of public assistance benefits in model 3 (AOR = 1.50, 95% CI: 1.09–2.08).

### 3.3. Outcomes by Race/Ethnicity and Experiences of Discrimination

In fully adjusted models stratified by mothers’ race/ethnicity, associations between EODs and the outcomes were largest among Latinx mothers (Table 4). EODs in at least one setting was associated with every outcome among Latinx mothers, while among Black mothers only HFI was associated with EODs in some settings and among white mothers none of the outcomes were statistically significantly associated with EODs in any setting. Among Latinx mothers, those who reported EODs had higher odds of HFI compared to those who reported no EODs in the same setting for EODs from police or in courts (AOR = 2.14, 95% CI: 1.33–3.45), in hiring and workplaces (AOR = 1.76, 95% CI: 1.21–2.56), and in school (AOR = 1.58, 95% CI: 1.05–2.36), but not in public places (AOR = 1.38, 95% CI: 0.97–1.98). Among Black mothers, odds of HFI were also elevated for EODs from police or in courts (AOR = 2.00, 95% CI: 1.32–2.96) and in hiring and the workplace (AOR = 1.46, 95% CI: 1.05–2.04), but not for EODs in public places or in school. Although not statistically significant, the estimate for the association between EODs from police or in courts and HFI was similar among non-Latinx white mothers as among mothers from the other groups (AOR = 1.83, 95% CI: 0.81–4.12).

EODs in school were associated with almost 80 percent higher odds of depressive symptoms among Latinx mothers (AOR = 1.79, 95% CI: 1.12–2.86). EODs in school were not associated with depressive symptoms among Black or white mothers, and EODs in other settings were not associated with depressive symptoms in any group. Among Latinx mothers only, poor physical health status was also associated with EODs in public places (AOR = 1.63, 95% CI: 1.12–2.38) and EODs in school (AOR = 1.96, 95% CI: 1.29–2.99).

## 4. Discussion

Lifetime experiences of racial and ethnic discrimination are associated with HFI, depressive symptoms, and fair/poor physical health for mothers of young children—in some cases, independently of ACEs. These results corroborate results found by Burke et al. [34], where lifetime discrimination in workplaces and schools was associated with the increased severity of HFI among a sample of 154 African American households. Our results were obtained with a significantly larger and more diverse sample, allowing us to demonstrate that the association between EODs and negative health outcomes persists despite participation in public benefit programs. Higher odds of poor health among mothers who experienced discrimination in public as well as higher odds of HFI among mothers who experienced discrimination from the police or courts and in the workplace were attenuated by adjustment for ACEs and public assistance but remained statistically significant. Our results also demonstrated that discrimination experienced in different settings has varying relationships with physical and mental health. For example, while discrimination experienced in public places was not associated with HFI, it was associated with poor physical health.

While in most cases adjustment for ACEs attenuated associations of EODs with the outcomes, suggesting the importance of accounting for other adversities when investigating the health impacts of EODs, the association between discrimination experienced in school and HFI and with physical health was strengthened by adjustment for ACEs. A plausible explanation for this particular change is that EODs in school may overlap with experiences of being bullied. Nearly 1 in 10 individuals with high exposures to ACEs also experience childhood bullying [35]. Victims of bullying have increased risk of poor health, dropping out of school, unstable employment, and living in poverty, independent of hardships and family dysfunction in childhood [36]. Thus, adjusting for ACEs may have captured the independent impacts of bullying rather than discrimination per se, which may explain the negative confounding phenomenon demonstrated in the results.

Although Black non-Latinx mothers reported the highest rates of discrimination, increases in odds of poor outcomes other than HFI associated with discrimination were statistically significant only for Latinx mothers. This may be due to the higher number of Latinx mothers in our sample, leading to greater ability to detect associations in this group. Or, it may be due to unmeasured factors in how Latinx mothers experience and react to discrimination which may be different from non-Latinx Black or white mothers. It should also be noted that the current study’s use of perceived experiences of discrimination, though important, does not fully capture total exposure to discrimination, which can go far beyond perceived experiences [37,38]. Experiences of institutional discrimination, such as residential segregation, differential access to societal goods and resources, and disparate sentencing laws, which were not accounted for in the study analyses, also adversely affect health and can compound the effects of perceived interpersonal discrimination.

Similarly, the lack of association between discrimination and outcomes for non-Latinx white mothers may reflect the smaller sample and lower rates of reported discrimination in this group, as well as societal and structural factors that shield non-Latinx white mothers from effects of discrimination. For example, while the majority of adults living in the U.S. are white, poor white families are not residentially concentrated in ways that Black and Latinx families are, due to residential segregation and historically racist policies that exploit or exclude communities of color [39,40]. Residential segregation leads to a “concentration of economic and social disadvantage and the absence of an infrastructure that promotes opportunity,” including: lower access to affordable, healthy foods, poorer housing quality, and increased exposure to environmental toxins, all of which are associated with worsened health and higher rates of premature death [5,41,42].

There are many plausible pathways between discrimination and HFI and health. Discrimination by police and courts has been linked to higher incarceration rates, which negatively impacts income, food security, and health [43]. Workplace discrimination based on race, ethnicity and gender can affect wages, job security, and ability to be promoted, which in turn affects income and mental health [44]. Finally, discrimination in schools is associated with differences in disciplinary actions that affect school performance which in turn affects mental health and income [45].

This study has a cross-sectional design, hence a temporal, causal link between discrimination and poor health cannot be made. Study results are also limited in that the mothers are from a single pediatric hospital; there was no regional variation to situate exposure to discrimination, health, and HFI within larger societal forces, such as housing segregation and disparate school funding. Also, health outcomes were self-reported rather than diagnosed by a health care provider, although the measures have been validated for a wide variety of settings. Recruiting participants through a hospital emergency department may introduce selection bias toward participants with poorer health overall, although in our study the mothers were at the emergency department to seek care for their children, not themselves, and the majority of mothers reported excellent or good physical health. Although fewer than 5 percent of mothers in the sample reported discrimination while getting medical care, it is possible that prior experiences of discrimination in health care settings might dissuade mothers from seeking care at the emergency department, leading to an underrepresentation in our sample of mothers who had experienced discrimination in this setting. Finally, due to the nature of our sample, our paper does not take into account the intersectional nature of gender and race/ethnicity to account for known ways in which the confluence of gender, racial and ethnic discrimination impact poor health and food insecurity [46,47,48,49,50].

Despite its limitations, this study contributes to the public health literature in several ways. First, it investigates the relationship between discrimination, HFI and health across multiple racial/ethnic groups using a large sample demonstrating that, while discrimination is harmful for everyone, it may be particularly harmful to Black and Latinx families. It also accounts for compounded adversities during childhood such as abuse, neglect, and household dysfunction captured by ACEs. Our study both answers the call for greater use of and adds to the evidence base for Critical Race Theory, which integrates self-reflection on methods and research questions, and thorough attention to dynamics of racism and poor health in order to get to the roots of poor health and health disparities [10]. Our study answers the call by bringing focused attention to discrimination in a variety of settings and assessing their impact on food insecurity and health among women and their young children. It adds to the evidence base by empirically demonstrating that the relationship between interpersonal discrimination and worsened health persists despite receipt of public benefits. In turn, this illuminates the mechanism(s) by which larger societal forces of racial and ethnic discrimination can contribute to poor health and adversity.

Our study’s finding that public assistance did not mitigate the relationship between discrimination and poor health, sheds light on how public benefits meant to address HFI and health may fall short. This may be for a variety of reasons rarely considered by researchers such as social position, differential treatment by the police and courts, in schools and workplaces, as well as potentially experiencing discrimination while shopping for food [51]. We also join others to frame these experiences with racial/ethnic discrimination as a form of violence exposure that is fundamentally associated with HFI, poor physical health and depressive symptoms [52]. This study highlights the severity of the associations between discrimination and health outcomes, above and beyond other trauma (such as ACEs) and in spite of programs intended to improve health, such as health insurance and receipt of public assistance. Future research should take into account the intersectional nature of discrimination to ensure that public health programs attend to issues of gender and gender expression while addressing racial and ethnic disparities.

## 5. Conclusions

The results from this study demonstrate that household food insecurity, depressive symptoms, and poor physical health are significantly associated with experiences of racial and ethnic discrimination by police and the courts, in public places, and in school and workplaces. These associations underscore the growing evidence that racial and ethnic discrimination is a major public health crisis. Public assistance programs and public health interventions intended to improve the health of populations will only be partially effective if we do not simultaneously work to reduce and eliminate the interpersonal, systemic and structural violence of racism and discrimination.

## Figures and Tables

**Table 1 ijerph-16-04369-t001:** Sample Characteristics (N = 1372) *.

Child mean age (months) ^ǂ^	19.7 (13.7)
Mother mean age (years) ^ǂ^	27.4 (5.8)
Race/Ethnicity	
Latinx	772 (56.3)
Black non-Latinx	447 (33.2)
White non-Latinx	106 (7.9)
Other	47 (3.4)
Mother’s region of birth	
United States and Puerto Rico	1176 (85.8)
Caribbean (Dominican Republic, Haiti, and Jamaica)	97 (7.1)
Mexico and Central America	54 (3.9)
Other	44 (3.2)
Marital Status	
Married or partnered	245 (17.9)
Not married or partnered	1123 (82.1)
Education	
Some high school or less	268 (19.5)
High school graduate	624 (45.5)
Technical school/college/masters	480 (35.0)
Currently employed	1138 (82.9)
Currently receive SNAP	1050 (76.6)
Currently receive WIC	975 (71.1)
Currently receive TANF	288 (21.1)
Currently receive housing subsidy	105 (8.8)
Health insurance	
Public insurance	1128 (82.2)
No insurance	117 (8.5)
Private insurance	123 (9.0)
Household food security	
Food secure	1134 (82.7)
Food insecure	238 (17.4)
Self-rated physical health	
Excellent or good	1075 (78.5)
Fair or poor	294 (21.5)
Experience depressive symptoms	237 (17.3)
Adverse Childhood Experience Score	
No ACEs	718 (52.3)
1–3 ACEs	489 (35.6)
≥4 ACEs	165 (12.0)

Abbreviations: SNAP, Supplemental Nutrition Assistance Program; WIC, Special Supplemental Nutrition Program for Women, Infants, and Children Women; TANF, Temporary Assistance for Needy Families; ACEs, Adverse Childhood Experiences. * Except where noted, values are frequencies with percentages in parentheses. Results may not add up to 100% due to rounding. ^ǂ^ Values are means with standard deviations in parentheses.

**Table 2 ijerph-16-04369-t002:** Experiences of Discrimination by Race/Ethnicity (N = 1372) ^ǂ^.

	All Mothers (N = 1372)	Latinx (n = 772)	Black Non-Latinx (n = 447)	White Non-Latinx (n = 106)	*p*-Value *
Setting of at Least One Experience of Discrimination	N (%)	n (%)	n (%)	n (%)	
From police or in the courts	130 (9.5)	76 (9.8)	84 (18.8)	11 (10.3)	**<0.0001**
On the street or in a public setting	373 (27.2)	161 (20.9)	186 (41.6)	20 (18.9)	**<0.0001**
Getting services in a store or restaurant	276 (20.1)	119 (15.4)	138 (28.3)	14 (13.2)	**<0.0001**
Getting hired or getting a job	221 (16.1)	103 (13.3)	99 (30.9)	15 (14.2)	**0.01**
At work	276 (20.1)	137 (17.7)	115 (25.7)	19 (17.9)	**0.02**
At school	268 (19.5)	128 (16.5)	113 (25.3)	24 (22.6)	**0.01**
Getting housing	85 (6.1)	39 (5.1)	39 (8.7)	6 (5.7)	0.09
Getting medical care	69 (5.0)	38 (4.9)	23 (5.1)	8 (7.5)	0.45
Applying for public assistance programs	94 (6.9)	16 (2.1)	7 (1.6)	7 (6.6)	0.23
Getting credit, bank loans, or a mortgage	58 (4.0)	25 (3.3)	30 (6.2)	3 (2.8)	**0.02**

EODs, Experiences of Discrimination. ^ǂ^ Mothers who indicated more than one race/ethnic identity or identified as Asian, Native American, or Other are not shown due to low sample sizes. * Tested by Fisher’s Exact Test. Bolded results are statistically significant (*p* < 0.05).

**Table 3 ijerph-16-04369-t003:** The Association between Experiences of Discrimination (EODs), HFI and Health Outcomes (N = 1372) *.

		Model 1 ^a^	Model 2 ^b^	Model 3 ^c^
		AOR (95% CI)	AOR (95% CI)	AOR (95% CI)
**Household food insecurity (HFI)**				
**EODs**	from police or in courts	**2.54 (1.82–3.53)**	**2.06 (1.45–2.91)**	**2.04 (1.44–2.89)**
	public places	**1.67 (1.28–2.12)**	1.31 (0.99–1.72)	1.31 (0.99–1.73)
	hiring and workplaces	**1.89 (1.44–2.48)**	**1.57 (1.18–2.10)**	**1.57 (1.18–2.11)**
	in school	**1.05 (1.01–1.10)**	**1.44 (1.06–1.96)**	**1.44 (1.06–1.96)**
**Maternal depressive symptoms**				
**EODs**	from police or in courts	**1.82 (1.24–2.66)**	1.23 (0.81–1.85)	1.22 (0.81–1.85)
	public places	**1.65 (1.16–2.33)**	1.26 (0.91–1.75)	1.28 (0.92–1.77)
	hiring and workplaces	**1.75 (1.28–2.39)**	1.22 (0.87–1.73)	1.24 (0.88–1.96)
	in school	**1.96 (1.30–2.97)**	1.39 (0.98–1.98)	1.40 (0.98–2.00)
**Poor physical health**				
**EODs**	from police or in courts	**1.47 (1.02–2.12)**	1.22 (0.83–1.79)	1.22 (0.83–1.79)
	public places	**1.59 (1.22–2.08)**	**1.36 (1.01–1.83)**	**1.37 (1.02–1.84)**
	hiring and workplaces	**1.36 (1.02–1.83)**	1.15 (0.84–1.58)	1.15 (0.84–1.58)
	in school	1.04 (0.99–1.09)	**1.95 (1.28–2.97)**	**1.50 (1.09–2.08)**

* Results are presented as adjusted odds ratios (AOR) with 95% confidence intervals in parentheses. Bolded results are statistically significant (*p* < 0.05). ^a^ Model 1 variables included maternal age, race/ethnicity, educational attainment, employment, Medicaid coverage, years in the United States (≥ 10 years in US) marital status, and household size. ^b^ Model 2 variables included all variables from model 1 and Adverse Childhood Experience (ACE) score. ^c^ Model 3 variables included all variables from model 2 and an indicator for current receipt of public benefits (housing subsidies, TANF, SNAP, or WIC).

**Table 4 ijerph-16-04369-t004:** The Association between Experiences of Discrimination (EODs) and Health Outcomes by Race/Ethnicity *^,^^ǂ^ (N = 1372).

		Latinx (n = 772)	Black Non-Latinx (n = 447)	White Non-Latinx (n = 106)
		AOR (95% CI)	AOR (95% CI)	AOR (95% CI)
**Household food insecurity (HFI)**				
**EODs**	from police or in courts	**2.14 (1.33–3.45)**	**2.00 (1.32–2.96)**	1.83 (0.81–4.12)
	public places	1.38 (0.97–1.98)	1.26 (0.92–1.73)	1.15 (0.63–2.10)
	hiring and workplaces	**1.76 (1.21–2.56)**	**1.46 (1.05–2.04)**	1.22 (0.65–2.27)
	in school	**1.58 (1.05–2.36)**	1.37 (0.97–1.93)	1.19 (0.61–2.31)
**Maternal depressive symptoms**				
**EODs**	from police or in courts	1.30 (0.74–2.29)	1.17 (0.73–1.89)	1.05 (0.41–2.73)
	public places	1.45 (0.95–2.22)	1.17 (0.81–1.70)	0.95 (0.47–1.92)
	hiring and workplaces	1.40 (0.89–2.19)	1.15 (0.78–1.69)	0.94 (0.46–1.96)
	in school	**1.79 (1.12–2.86)**	1.21 (0.81–1.81)	0.95 (0.47–1.92)
**Poor physical health**				
**EODs**	from police or in courts	1.38 (0.82–2.30)	1.12 (0.71–1.77)	0.91 (0.36–2.30)
	public places	**1.63 (1.12–2.38)**	1.19 (0.84–1.69)	0.87 (0.45–1.71)
	hiring and workplaces	1.38 (0.92–2.07)	1.01 (0.70–1.47)	0.74 (0.36–1.51)
	in school	**1.96 (1.29–2.99)**	1.25 (0.85–1.84)	0.80 (0.38–1.69)

* Results are presented as adjusted odds ratios (AOR) with 95% confidence intervals in parentheses. Models are adjusted for maternal age, educational attainment, employment, insurance coverage, years in the United States (≥ 10 years in US) and marital status, household size, Adverse Childhood Experience (ACE) score, and an indicator for current receipt of public benefits (housing subsidies, TANF, SNAP, or WIC). ^ǂ^ Mothers who indicated more than one race/ethnic identity or identified as Asian, Native American, or Other are not shown due to low sample sizes. Bolded results are statistically significant (*p* < 0.05).

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
