# Peer review of "Experiences of Racial and Ethnic Discrimination Are Associated with Food Insecurity and Poor Health"

_ijerph, 2019, doi:10.3390/ijerph16224369_

Round 1
Reviewer 1 Report
Please see attached.

Author Response
Thank you so much for the thoughtful/insightful review and commentary. Your recommendations make this a better manuscript. All changes are highlighted in yellow in the text version of the manuscript.
Reviewer Comments & Author Responses:
Reviewer #1
Article Summary: This study examines the influence of experiences of discrimination on food security and health outcomes (maternal depression and physical health) accounting for adverse childhood experiences and food assistance.
My perspective: I am a public health researcher focusing on food security, however in a very different setting. Methodologically, I have experience using logistic regression to model exposures with several outcomes accounting for mediators, moderators, and interactions. Strengths: The literature review provides a good summary of the literature in this area and how the present analysis contributes.
The study design and analytic strategy are strong and account for multiple adversities (HFI, EOD, and ACES) that may increase disparities. Tables are clearly organized and present all relevant information for the reader.
--thank you for these thoughtful responses.
INTRODUCTION: a. In the Introduction, consider adding the framework of critical race theory to orient the reader and provide a theoretical lens through which to view the study. You make a strong case for the importance of critical race theory in health research and after hearing that case, I wanted to go back and re-read the manuscript with that in mind. Other readers may also like to have that framework in mind while reading--Thank you for this recommendation. We agree it is stronger if we describe critical race theory up front. We have added text to briefly introduce the Critical Race Theory framework (see lines 46-57).
METHODS: In the Methods section, a bit more detail on the methods of data collection in this manuscript would be useful to the reader. Specifically, I think the following information would be helpful: Inclusion/exclusion criteria How was critical condition defined/determined by the trained interviewer?iii. How were decisions made about who to approach, who not to approach?
How often was data collected, every day for over 2 years, certain days of the week, times of the day, times of the year? Response rate, how many mothers were approached, how many were eligible, how many agreed to participate What medical records data were obtained, were they used in this analysis?Thank you for the opportunity to clarify our study methods. In the highlighted sections, you will see that we have clarified to indicate that: i) mothers were taken from a convenience sample eligible participants were parents (primary caregivers) of children under 4yrs who were not in critical condition; ii) critical condition was defined by triage staff assignment or any patients admitted due to assault injuries; iii) parents were approached when they were unoccupied while waiting for a nurse or physician; iv) data was collected during regular business hours.
Please also add the validity/reliability data points for the HFSSM and ACES for the reader, rather than stating high or strongThank you for this recommendation. We have added the numeric evidence of validity/reliability
Finally, the 9-item EOD was described as slightly modified, please describe how this was modified.Thank you for this recommendation. We clarified that we added a single question to capture the context of applying for public assistance.
RESULTS: a. In the Results section, the association strengthens for EOD in school with the outcome HFI (section 3.2) as you account for ACES and food assistance (Table 3). This is not described in the results or discussed. What do you think is going on in school settings with food security and discrimination? b. Also, please describe how missing data for your study measures were handled.Comment a): HFI has been added to the discussion (lines 273-276).
Comment b): we state that of the initial sample of 1,707 participants, 1,643 completed the survey in full. The 1,372 mothers in the analytic sample were drawn from the 1,643 parents who completed the survey in full and there were no skipped responses. Thus, there were no missing data. We add this sentence on line 93.
Reviewer 2 Report
Thank you for the opportunity to review this important topical paper. This is an interesting subject which appears to have been under-researched previously. The paper is exceptionally well written and benefits from an excellent sample size.
The abstract is concise and purposefully written. The background section is succinct and informative. The authors have presided over an ethically-compliant study. The study population is well discussed.
I would suggest (line 41) explaining the HFSSM scale and cut-off points for household food (in)security for absolute clarity for the reader.
There are clear attempts to describe and discuss the limitations of the study.
The authors have written a very valuable conclusion as a contribution to the discipline and recommendations for interventions in the future (ie) merits of investigating co-existence of HFI and EOD, for maximum public health impact.
Minor additional proofreading required:
Line 38 - should read as 'disproportionately' Line 156 - insert the missing word 'to' - 'in order to descriptively assess...' Line 162/3 - Line return needed after line 162 Line 214 - insert the missing word 'to' - 'further to adjusting...'
Author Response
Thank you so much for the thoughtful/insightful review and commentary. Your recommendations make this a better manuscript. All changes are highlighted in yellow in the text version of the manuscript.
Reviewer #2
Thank you for the opportunity to review this important topical paper. This is an interesting subject which appears to have been under-researched previously. The paper is exceptionally well written and benefits from an excellent sample size.
The abstract is concise and purposefully written. The background section is succinct and informative. The authors have presided over an ethically-compliant study. The study population is well discussed.
There are clear attempts to describe and discuss the limitations of the study.
The authors have written a very valuable conclusion as a contribution to the discipline and recommendations for interventions in the future (ie) merits of investigating co-existence of HFI and EOD, for maximum public health impact.
Thank you for these thoughtful comments.
I would suggest (line 41) explaining the HFSSM scale and cut-off points for household food (in)security for absolute clarity for the reader.
Thank you for this opportunity to make this more clear. Our primary interest is to examine the impacts of exposure to discrimination at a broad level, but we recognize that there are important subtleties in HFSSM cutpoints. We address the Food insecurity cut points in the methods section. For brevity, we chose to keep the explanation of the HFSSM in the measures portion of the methods section.
Reviewer 3 Report
It was a pleasure to read this well written, thoughtful, and useful paper. I think it can be published without modification. Although the data certainly does have limitations (which the authors document adequately in their discussion), the paper presents powerful (though unsurprising) evidence for the relationship between EOD and factors such as HFI and depression. To have these disaggregated by racial/ethnic identity is particularly important. For me, the result that EOD had greater impact for Latinx than black respondents was somewhat surprising. That opens a path to more research as does the authors' comment on intersectionality in the discussion.
The authors say little about causality here, and that is good. It seems to me that, ultimately causality must lie in structural-historical factors that undergird American society. At the same time, it may be useful to look at EOD, food insecurity, poor health, and depression more specifically as a syndemic.
Thanks much for the opportunity to review this outstanding paper.
Author Response
Reviewer #3
It was a pleasure to read this well written, thoughtful, and useful paper. I think it can be published without modification. Although the data certainly does have limitations (which the authors document adequately in their discussion), the paper presents powerful (though unsurprising) evidence for the relationship between EOD and factors such as HFI and depression. To have these disaggregated by racial/ethnic identity is particularly important. For me, the result that EOD had greater impact for Latinx than black respondents was somewhat surprising. That opens a path to more research as does the authors' comment on intersectionality in the discussion.
The authors say little about causality here, and that is good. It seems to me that, ultimately causality must lie in structural-historical factors that undergird American society. At the same time, it may be useful to look at EOD, food insecurity, poor health, and depression more specifically as a syndemic. Thanks much for the opportunity to review this outstanding paper
Thank you for these very thoughtful comments. We like the concept of “syndemic” and we are grateful that this concept came through in the paper. We hope our paper stimulates further research.